# Modelling Green VTOL Concept Designs for Reliability and Efficiency

**Julian David Booker \*** , **Caius Patel and Phillip Mellor**

Electrical Energy Management Research Group, Faculty of Engineering, University of Bristol,
Bristol BS8 1TR, UK; gd18875@bristol.ac.uk (C.P.); p.h.mellor@bristol.ac.uk (P.M.)
\* Correspondence: j.d.booker@bristol.ac.uk

**Abstract:** All-electric and hybrid-electric aircraft are a future transport goal and a possible 'green' solution to increasing climate-related pressures for aviation. Ensuring the safety of passengers is of high importance, informed through appropriate reliability predictions to satisfy emerging flight certification requirements. This paper introduces another important consideration related to redundancy offered by multiplex electric motors, a maturing technology which could help electric aircraft manufacturers meet the high reliability targets being set. A concept design methodology is overviewed involving a symbolic representation of aircraft and block modelling of two important figures of merit, reliability, and efficiency, supported by data. This leads to a comparative study of green aircraft configurations indicating which have the most potential now, and in the future. Two main case studies are then presented: an electric tail rotor retrofitted to an existing turbine powered helicopter (hybrid) and an eVTOL aircraft (all-electric), demonstrating the impact of multiplex level and number of propulsion channels on meeting target reliabilities. The paper closes with a summary of the important contribution to be made by multiplex electric machines, well as the advancements necessary for green VTOL aircraft sub-systems, e.g., power control unit and batteries, to improve reliability predictions and safety further.

**Keywords:** green VTOL; reliability; modelling; redundancy; multiplex

## 1. Introduction

Green VTOL aircraft are characterized by electric motors directly driving rotors, fans, and propellers, providing lift and thrust. The motors may be powered either by batteries (termed all-electric or eVTOL aircraft), or the electric motors can be powered by generators connected to more conventional turbines or internal combustion engines, in combination with batteries (termed hybrid-electric) [1]. A key difference from conventional aircraft and what can be termed, green VTOLS, is that the electric propulsion is separated from the engine, thereby new aircraft configurations become feasible [2]. Electric propulsion can be used continuously throughout flight or for a part of a mission, e.g., boosting power for take-off or landing [3]. Though more complex, hybrid-electric aircraft also have the potential to increase the mission range over all-electric designs [4–6]. Various aircraft designs and demonstrators have been proposed at various levels of technical readiness [7–10], with commercial flights expected between 2022 and 2025 [11–13].

Through the use of electric motors for propulsion, and the corresponding high use of electricity rather than relying completely on on-board hydrocarbon-based fuels, green VTOLs aim to produce lower or even zero emissions to help meet the challenging emissions reduction targets set by, for example, the Advisory Council for Aeronautics Research in Europe [14,15]. There are additional benefits of electric propulsion proposed, including: reduced vibration and noise, increased efficiency, improved energy consumption, improved maintainability, reduced operating costs, lower mass, and aerodynamic improvements [3,8,16]. A key driver for green VTOLS to deliver these potential improvements is the development of electric propulsion systems [17]. Improvements in electric

machine power density and efficiency, energy storage capacity and efficiency of batteries, and reliability of power converter technologies are all required to make electric propulsion for aircraft a more realistic proposition [1,18,19].

The use of electrical machines for propulsion also offers additional improvements related to redundancy and fault tolerance, not typically offered by conventional aircraft architectures, to improve reliability primarily [2,8,13,16,18,20–22]. Fault tolerance is the ability for a system to still maintain the desired reliability despite failure of any component or sub-system. Conventionally, a number of the same safety critical sub-systems can be arranged in parallel to increase overall system reliability. In terms of an electrical motor, fault tolerance can be achieved by using concentrated windings partitioned on the stator, powered and controlled independently, creating a multiplex machine with a single output shaft—effectively creating a parallel configuration and therefore maintaining high reliability of the output from the motor. The result of this partitioning (duplex, triplex, quadruplex redundancy, etc.) is electrical, magnetic, and thermal insulation between phases [7,23,24], achieving high reliability potential for safety critical applications, such as aircraft [25–27]. If failure in one of the winding partitions occurs, there is only a degradation in output power, not a complete loss, allowing continued lift and thrust for controlling the aircraft and avoiding catastrophic consequences. Therefore, designing in redundancy will have a significant impact on the availability of thrust following failure of an electrical component [3]. The aerospace industry is already well-aware of multiplex systems for safety, e.g., flight control systems are based on triplex or quadruplex redundancy [28]. Though there are potential disadvantages, such as increasing the mass, cost, thermal management challenges, increased health monitoring requirements, and potential for increased overall complexity [21,26], the adoption of fault tolerant electrical machines is an attractive solution to meet safety and power availability targets for aerospace applications [29].

Various types of redundancy are implemented in practice. The Bell Boeing V-22 Osprey tiltrotor provides redundancy by coupling the power from both main turbines through a central gearbox, meaning if one of the two turbines fails, power is still available to both propellers, though overall power is reduced by 50% [30]. However, a considerable number of components link the turbines and the propellers through mechanical transmissions, raising reliability concerns [31], and significant failures in rotorcraft can be attributable to main engine and transmission failure [32]. Moving to direct electrical motor drive of propellers, the all-electric VoloCity, is an 18-rotor distributed propulsor design [33], providing redundancy through high number of the same motor. Similarly, the eHang autonomous aircraft has two motors each with its own propeller coaxially arranged at up to eight locations on the aircraft, providing 16 independent propulsion units in total [34]. More recently, however, we have seen redundancy within motors being applied. For example, the all-electric eFlyer 800 fixed wing aircraft from Bye Aerospace proposes two important safety features: two wing-mounted electric motors, each with dual redundant motor windings powered by quad-redundant battery packs [35]. Finally, an electric tail rotor for an existing helicopter was ground tested recently by Leonardo, which uses a quadruplex motor, demonstrating little increase in mass over a conventional simplex machine of similar rating whilst providing a high degree of fault tolerance [36]. Confirmation that there was no appreciable mass penalty in producing a multiplex motor compared to a simplex version was an important outcome, especially for an aerospace propulsion application, and contradicts previous assumptions [21].

Generally, there are two schools of thought emerging: multiple propulsion units or multiplex machines to reduce the consequence of a single failure and therefore improve reliability, power availability, and aircraft safety [31]. There is no reason why a design cannot incorporate both redundancy approaches, and their level of adoption and integration is highly dependent on the reliability targets to be met. However, the type and level of redundancy will also need to be related to the critical failures, hazards, and faults, which through complete loss or degradation of propulsion, relate to the consequences of failure, stability, and control of the designed aircraft.

This paper will present a systematic method for assessing reliability and efficiency figures of merit of different green VTOL designs incorporating redundancy, supported with data to compare different configurations quantitatively and against targets. This paper will first introduce the methodology, including an approach to help visualize aircraft architectures using standard symbolic representation and block diagrams, before presenting a comparative study of the known hybrid and fully electric aircraft architectures from the literature, with variations incorporating redundancy to compare reliability and efficiency. The method will then be demonstrated in two case studies, before drawing conclusions and areas for further research.

## 2. Methodology

### 2.1. Symbolic Representation

Potentially, there are many different aircraft architectures that are conceivable at the concept level for green VTOLs. It is essential to have a standard way of creating the designs symbolically, as this will provide a visual reference of all concepts on a common basis. This symbolic representation also shows how the different sub-systems are connected, making the creation of Block Diagrams (for reliability and efficiency modelling later) more systematic. The American Institute of Aeronautics and Astronautics (AIAA) have developed a set of symbols for modelling hybrid-electric aircraft systems, which will be adapted for use here [37]. Figure 1 shows a selection of commonly used symbols and data for reliability and efficiency of components and sub-systems can also be sourced from the literature; reliability and efficiency are the two key figures of merit introduced later in the methodology. However, adaptation and addition of some symbols is required. A PCU (Power Control Unit) symbol is introduced to reduce the complexity of the power control system, combining inverters and rectifiers, etc. A PCU symbol must also be connected in advance of each electrical motor wherever specified in the concept system.

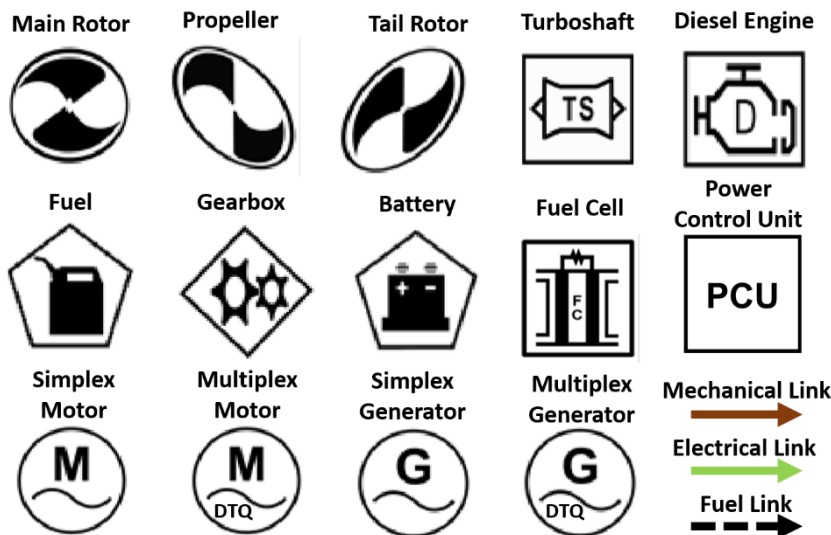

**Figure 1.** Selected AIAA Symbols Used for System Representation (adapted from [37] and reprinted by permission of the American Institute of Aeronautics and Astronautics, Inc.).

The current industry standard is based around three phase electrical drive technologies. For example, fully integrated power modules are commercially available as complete three-phase units and a closely coupled common DC link filter capacitor. However, it is recognized that other degrees of freedom are possible. A true fault tolerant electrical machine design provides electromagnetic and a degree of thermal independence between the phase windings, thus with appropriate adjustment to the pole-slot design, other groupings than three-phase groupings would be possible, e.g., five phases, six phases, or others. The data capture has shown the PCU, which includes the DC link capacitor, EMI filtering, gate drive,

switching, and current control elements alongside the power semiconductor devices, to be the least reliable component in the electrical energy supply chain. Here, for simplicity, it has been assumed that any failure within the power electric unit would make the complete unit inoperable and the fault-tolerant architecture is configured to have a benign effect on or be isolated from the other connected elements. In the case of a multiphase drive, depending on the nature and criticality of the fault within the PCU, it can be possible for the PCU and connected machine phases to continue to function at reduced capacity following a failure and therefore could be beneficial in reducing the number of independent power channels needed. The analysis of this would require greater granularity of reliability data, in understanding failures at a PCU sub-element level, and will be the subject of future work.

Two sets of motor and generator symbols are also adapted for use; one set for the standard simplex topology, and the other set for multiplex machines, where the multiplex versions of the symbols have the letters 'DTQ', where D = Duplex, T = Triplex, and Q = Quadruplex topologies. For example, the Electric Tail Rotor (ETR) motor mentioned earlier, is a Quadruplex machine, and would use the letter Q for the motor. These symbols are easily stored within interactive digital whiteboards, where they can be arranged to create the system designs desired with the necessary mechanical, electrical, and fuel links connecting them.

*2.2. Figures of Merit*

The comparison of each concept system is conducted using two simple overall figures of merit; reliability and efficiency. Reliability can be defined as the ability of a system or component to perform its required functions under stated conditions for a specified period, whereas efficiency is the ratio of the energy output to the energy input in each system. In the development of many electro-mechanical systems, reliability can be regarded as a key technical driver, also related to safety, and efficiency as a key economic driver, also related to environmental performance.

Figure 2 shows a Pareto Chart combining the important figures of merit from six case studies in electro-mechanical system design [38], with the prominence of reliability (combined with safety) and efficiency indicated. Robustness and specific geometric/performance figures of merit are difficult to quantify at system conceptualization stages, and do not lend themselves to high level system modelling interpretation. However, the additional benefit of using reliability and efficiency is that their formulation is easily quantified from zero to one, where zero is related to no reliability/efficiency and one is complete reliability/efficiency. In reality, unity for reliability and efficiency is not achievable (through the laws of physics), and to allocate zero reliability or efficiency to any component or sub-system is nonsensical, but these figures of merit provide simple quantifiable measures from which decisions about the system viability can be made, both technically and economically. Although no targets for efficiency are proposed for a system, as will be introduced for reliability later, obviously this value should be as high as possible, i.e., $\eta_{sys} \rightarrow 1$.

Therefore, it is expected that different concept designs will achieve different reliability and efficiency levels somewhere between a number greater than zero and less than unity, which can be compared to each other or target values. These two figures of merit are also considered independent of each other and not related to scaling or power distribution in the system conceived. Therefore, a system of components and sub-systems each with an independent reliability and efficiency values can be used to quantify the overall system reliability and efficiency. The difficulty remaining is to apportion values of reliability and efficiency to each component and sub-system, as well as formulate the system mathematical models representing the configurations of components and sub-systems in an ideal way, which will be discussed next.

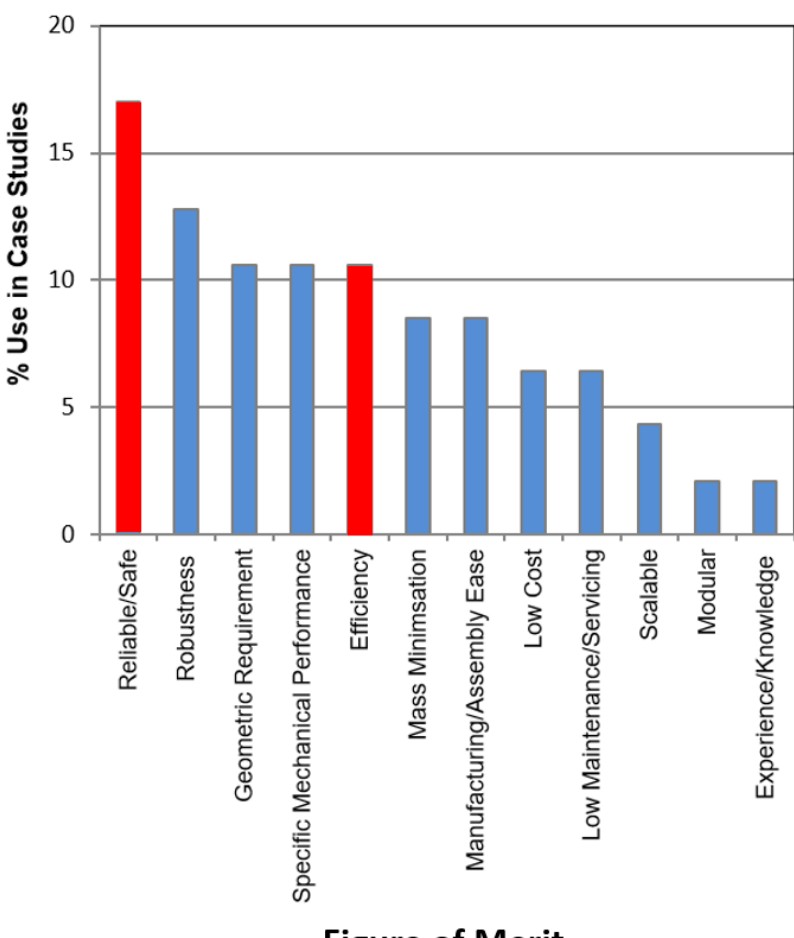

**Figure 2.** Pareto Chart showing Figure of Merit Use Across Electro-mechanical Systems Case Studies (adapted from [38]).

### 2.3. Block Modelling

The use of block models (or block diagrams) is a standard way of graphically representing a system made up of components and sub-systems using simple blocks connected by lines representing their relationships [39,40]. The component and sub-system blocks each represent the component and sub-system symbols, as created using the AIAA symbolic representation, though their configuration is dictated in essentially two ways only: in series or in parallel. Using the block model approach, the total reliability and efficiency values can be propagated, usually from left to right, when the data is known for each component and sub-system and whether the configurations are in series or in parallel. This then provides the next level of the system representation for mathematical modelling and prediction of the two figures of merit: Reliability (R) and Efficiency (η). The block models and associated equations for reliability and efficiency for both in series and in parallel configurations are shown in Figures 3 and 4, respectively. A system design will combine, as necessary, in series and in parallel configurations of components and sub-systems, with the major utility of in parallel configurations being for modelling multiplex approaches using redundancy to increase reliability of the overall system.

### 2.4. Reliability and Efficiency Data

Data for reliability and efficiency has been sourced from the relevant literature for all common components and sub-systems used in hybrid electric systems, in line with the AIAA symbols used. The complete set of data obtained is summarized in Table 1, together with references for the public domain sources. The original data is usually presented in the form of probability of failure, P, seldom a value of reliability, R, and has been converted

through the relationship, R = 1 − P. Two sets of data are actually shown: one set represents current values, or state-of-the-art, and one set is for future data applied to key electrical sub-systems, representing the best estimates from the available literature projected for the next 10 to 20 years, and supported by the authors' experience. Some of the current data for reliability is rotorcraft specific or has operational conditions applied, but all data must be used with caution in an absolute sense, as there maybe large variations depending on supplier, operating environment and application domain. Updating the data sourced with supplier data (where available) would improve confidence in the system level predictions made, though reliability data is very sensitive commercial information usually, and is difficult to obtain.

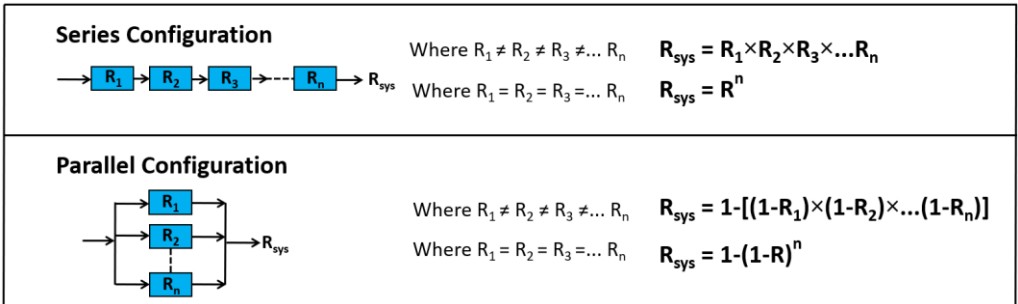

**Figure 3.** Block Models and Equations for Reliability.

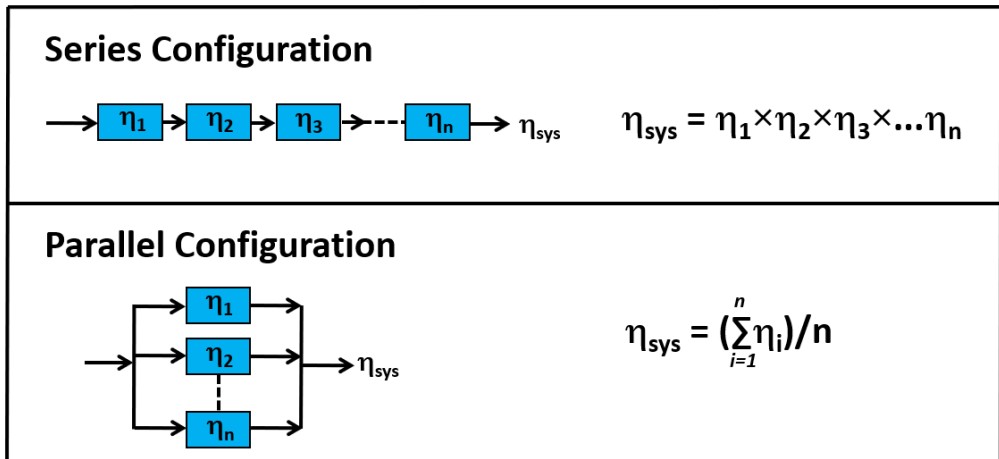

**Figure 4.** Block Models and Equations for Efficiency.

**Table 1.** Reliability and Efficiency Data for Selected Components and Sub-systems (* current and select future values, with future values shown as projections in bold).

| Component/Sub-System | Reliability (R) * | Efficiency (η) * |
|---|---|---|
| Main Rotor Gearbox ($x_{GG}$) | 0.9994 [41] | 0.94 [42] |
| Bevel Gearbox ($x_{GB}$) | 0.99995 [41] | 0.97 [43] |
| Multiplier/Reducer Gearbox ($x_{GM}$) | 0.9978 [41] | 0.94 [43] |
| Power Control Unit ($x_{PCU}$) | 0.84 [26] →**0.9** | 0.97 [44] →**0.98** |
| Turboshaft ($x_T$) | 0.9994 [41] | 0.3 [45] |
| Battery ($x_B$) | 0.82 [46] →**0.9** | 0.9 [47] →**0.99** |
| Fuel Cell ($x_F$) | 0.95 [48] →**0.98** | 0.65 [47] →**0.83** |

**Table 1.** *Cont.*

| Component/Sub-System | Reliability (R) * | Efficiency (η) * |
|---|---|---|
| Motor/Generator ($x_M/x_{GEN}$) | 0.9998 [41] →**0.99999** | 0.92 [44] →**0.99** |
| Diesel Engines (~200 kW) ($x_D$) | 0.78 [41] | 0.40 [49] |
| Shafting ($x_S$) | 0.9997 [41] | ~0.99 |
| Power Cables—Propulsion ($x_{CP}$) | 0.9992 [41] | 0.99 [50] |
| Power Cables—Batteries ($x_{CB}$) | 0.999998 [41] | 0.99 [50] |
| Propeller ($x_{PROP}$) | 0.9985 [41] | 0.87 [51] |

*2.5. Reliability Targets*

Green VTOLs may present new aircraft design opportunities, but they also present flight safety certification challenges, as unconventional solutions will be adopted that do not necessarily lend themselves to current certification requirements [9,52]. Furthermore, certification for proposed pilotless, autonomous aircraft carrying paying passengers over densely populated urbans areas will almost certainly set the bar even higher [1] and it has also been argued that eVTOLs will need to be much safer than current rotorcraft if they are to be successful economically [53]. Therefore, emerging guidance from the European Aviation Safety Agency (EASA) suggests that safety cases for flight certification will need to demonstrate that no single failure has catastrophic consequences on the aircraft and recommends the use of a risk assessment method, such as Failure Mode and Effects Analysis (FMEA) [54] in order to eliminate risks; a direction shared by researchers in the area [55,56]. FMEA, and more favored in aerospace, the related method Failure Mode, Effects and Criticality Analysis (FMECA), is traditionally applied to aircraft equipment to also help analyze reliability [57]. FMEA and FMECA consider not only the consequences of failure and their severity, but also the likely occurrence of the failure taking place, i.e., its probability of failure, or equivalently, its reliability. These are the two elements of risk: Occurrence (O), or how many times do we expect the failure to occur?; and Severity (S), what are the consequences of failure on the customer or environment? Therefore, the methodology presented here will also need to include setting reliability targets related to occurrence, aligned with severity of failure consequences, whilst being aligned to future certification targets.

The basic FMECA method [58] has been adapted for use on green VTOLS and a set of definitions for Severity (S) have been developed, as shown in Table 2, where the consequences of failure on a five-point scale range from insignificant (S = 1) to catastrophic (S = 5). Some potential failure conditions relate to both conventional and new green VTOL aircraft alike, such as the weather, terrain, and airborne bodies, e.g., bird strikes, which may incapacitate a propulsion unit completely through damage to the propeller or fan, though statistically, strikes on the fuselage are more likely. From air accident reports, the loss of a motor in a multi-rotor drone usually results in loss of control and subsequently, it will crash to the ground [59,60]. A lack of redundancy in a drone's control system also contributed to a recent catastrophic failure [61] it is noted. Therefore, there are many potential failures that will also relate to the electrical systems, e.g., battery, PCU, motors, etc. [62]. With respect to major green VTOL sub-systems, Table 3 shows a classification of the different failure modes that could be experienced in service, aligned with the Severity (S) ratings, with the effects of the failure described in this table [63]. From the twelve failure modes reported here (this is by no means an exhaustive list), nine are in the severe to catastrophic category, with no failure modes allocated low severity ratings. This indicates that if the severity of consequence cannot be reduced, which is typically the case due to application domain and costs, the occurrence of these failures needs to be very low indeed to maintain an overall low risk, therefore, the reliability would need to be very high.

**Table 2.** FMECA Severity (S) Ratings for Green VTOLs.

| Severity | Description | Rating (S) |
|---|---|---|
| Catastrophic | Sudden loss of primary sub-system function with direct safety implications and effects on green VTOL system | 5 |
| Severe | Sudden loss of primary sub-system function with potential safety implications on green VTOL system and secondary sub-system failures | 4 |
| Major | Gradual failure of sub-systems, which are operable but at reduced level of performance, directly impacting functionality—no immediate effect on green VTOL system | 3 |
| Minor | Failure resulting in minor implications to performance in sub-system functionality, with no direct safety implication on green VTOL system | 2 |
| Insignificant | No failure implications on sub-system performance, functionality or safety on green VTOL system | 1 |

**Table 3.** FMECA Severity (S) Ratings for Selected Green VTOLs Failure Modes and Effects.

| System Element | Failure Mode | Severity (S) | Effects of Failure |
|---|---|---|---|
| Propellor Failure | Bird Strike | 5 | Loss of lift from propellor resulting in loss of aircraft stability |
| | Structural Failure | 5 | Loss of lift from propellor resulting in loss of aircraft stability |
| Motor Failure | Winding Failure | 4 | Proportional loss of power and subsequent lift from propeller. Remaining components under more stress. |
| | Overheating | 3 | Loss of efficiency and performance. Possible further impacts upon components. |
| | Bird Strike | 5 | Complete loss of power, loss of lift. |
| | Overloading | 4 | Higher stresses upon components, causing further failures. |
| PCU Failure | Short Circuit | 4 | Loss of power from motor, cascade effect. More strain on other components. |
| | Overheating | 3 | Increased failure due to degradation. Possible further impacts. |
| Energy Storage | Puncture | 5 | Battery fire, considerable further failure within power system. Loss of power, loss of lift. |
| | Thermal Runaway | 4 | Loss of power, possible further battery failures. |
| | Overloading | 3 | Higher stress on batteries, could cause further degradation |
| | Short Circuit | 4 | Loss of power, provided. Possible thermal runaway. |

A decision must be made as to the reliability target appropriate for any application [64]. Research into the effects of non-conformance and associated costs of failure found that an area of acceptable design can be defined for a component characteristic on a graph of failure occurrence versus severity of consequence. The result was a Reliability Map, previously developed for general engineering [65], and provides target reliabilities based on FMECA ratings for Severity (S), as shown in Figure 5. The map includes areas associated with acceptable design (this limit set at a value of 1% of the total failure cost), unacceptable, conservative and overdesign. The overdesign area is probably not as important as the limiting failure probability for a particular severity rating but does identify possible wasteful and costly designs.

Various authors have presented target probability of failures ranging from $10^{-3}$ to $10^{-9}$ for various applications and conditions [66–68]. These values fit in well with the Reliability Map proposed, recognizing that as failures get more severe, they cost more, and so the objective must be to reduce the occurrence or increase reliability. Note that the Occurrence (O) rating is not shown on the Reliability Map but has been replaced by

the probability of failure and corresponding reliability targets. However, absolute values for these targets may be unrealistic [69] and therefore they should therefore be a central measure bounded by some range that spans a space of credibility, never a point value because of the underlying uncertainty in the data and models used [70] and the difficulty in verifying them at the concept stages.

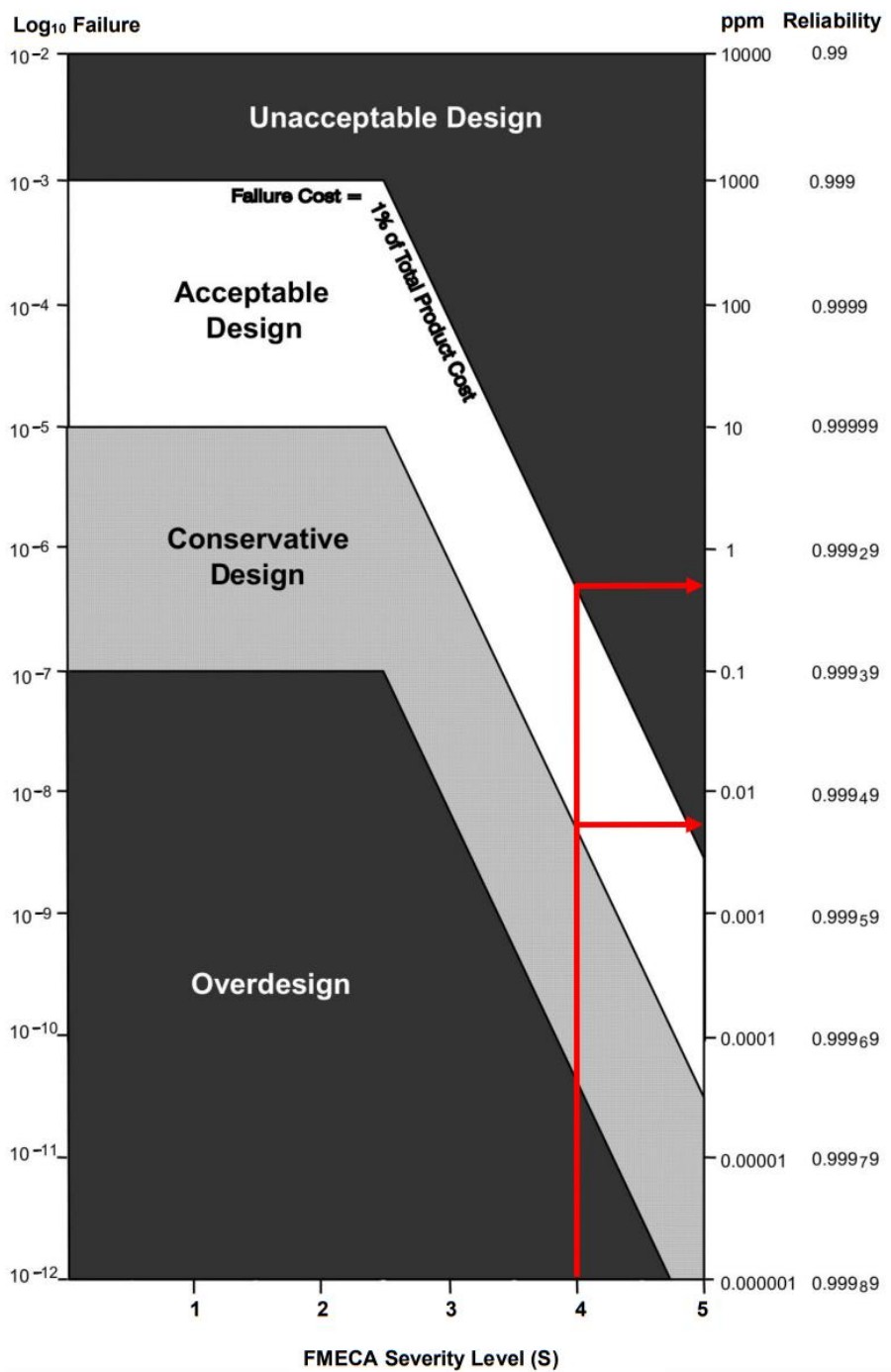

**Figure 5.** Reliability Target Map Showing Ranges for Different Severity (S) Ratings (adapted from [65]).

Reflecting on the targets set by EASA [71] for small VTOL aircraft, probabilities of failure range from $\leq 10^{-3}$ for minor to $\leq 10^{-9}$ for catastrophic severities. In addition, EASA [54] state that a target of $\leq 10^{-7}$ for lift/thrust systems should be used, which means R $\geq$ 0.9999999 must be demonstrated. This is interpreted as any single channel

of propulsion specified in the design should attain at least this level of reliability. Some researchers have gone as far to suggest that failure probabilities for green VTOLs should generally be targeted at $10^{-9}$ [53]. Whilst safety is of course a primary goal, and reliability and risk assessments can help direct resource and effort in order that components and sub-systems are of high reliability, targets set at a single high threshold do not take failure consequence into account and may lead to overdesign, mass penalties, and, therefore, increased power demand for lift, leading to overall system viability concerns. In practice, for a given consequence of failure and allocated Severity Rating (S), if the target range is not met with a particular preferred aircraft architecture, then adding in redundancy is recommended, e.g., parallel channels of power, control and propulsion, multiplex motors and generators, etc., which will be considered in the context of a range of single-channel propulsion systems next.

## 3. Single-Propulsion Channel Study

Seven single-channel propulsion configurations have been derived from the available information in the literature for green VTOLs [4,19,56,72–78]. The configurations, as basic representations, are shown in Figure 6, together with a generic name describing the configuration, with all including motors and generators of the simplex type. For the study discussed next, variations on these basic configurations also include the use of quadruplex motors and the use of current and future data for reliability and efficiency of components and sub-systems, as provided in Table 1. For example, for the category [A] All Electric Battery configuration, A1 = simplex motor using current data, A2 = quadruplex motor using current data, and A3 = quadruplex motor using future data. For each configuration, a set of assumptions is also made, e.g., for [D1] Series Hybrid Direct Drive, the generator (high speed) is directly driven by the turboshaft, both power sources are used simultaneously, i.e., boost mode (for take-off and landing purposes) and an additional benefit of battery charging capability is possible with this configuration. For all quadruplex motor configurations variants, there will also be four independent power channels with their own PCU, as discussed earlier. The merits and demerits of each of these configurations, as well as these assumptions, will not be explored further here, and the reader is referred to the various reference sources for further information. A judgement of the merit and demerit of each configuration will simply be made next based on only the predicted system reliabilities and efficiencies, termed $R_{sys}$ and $\eta_{sys}$, respectively, where the system is the single channel of propulsion.

An example calculation for the system reliability and efficiency is shown in Figure 7 for D1 using block models populated with the appropriate component and sub-system reliability values from Table 1. It is evident from the block models the level of redundancy provided when the D1 configuration when used in boost mode, which of course, is not the primary flight mode or a viable VTOL type aircraft with a single channel of propulsion; however, this first study attempts to compare each known configuration on a common basis. Also shown in Figure 7 is the final calculation for the equivalent block model on the left when series calculations for reliability and efficiency have been undertaken to the right-hand side, for clarity. Irrespective of the final reliability and efficiency values calculated, these will now be compared in a relative format for all basic configurations using simplex motors and current values and their variants using multiplex motors and future values.

The results for all configurations and their variants are shown in Figure 8 in a Pareto chart format, ranking those configurations with the highest reliability from left to right, together with their associated efficiencies. Though final predictions are not provided as absolute values, and graphically, several configurations appear to achieve reliabilities approaching unity, this is deceptive, and the highest reliability calculated was for B3, $R_{sys}$ = 0.998, or $P_{sys} \approx 10^{-3}$. This is substantially less than the target specified by EASA for lift/thrust systems [71] of at least $P_{sys} = 10^{-7}$. It is recalled that from the Reliability Map provided in Figure 5, a probability of failure of $10^{-3}$ was the very minimum acceptable for

non-safety critical applications. This shortfall, even for a simplified set of possible solutions for all and hybrid electric aircraft configurations prompts the question how, in practice, will manufacturers of new green VTOLs be able to demonstrate to regulators that they meet target reliabilities for flight certification purposes given the lack of confidence and access to accurate data for the reliability of components and subs-systems, as well as possible inaccuracies and assumptions in modelling the systems they conceive.

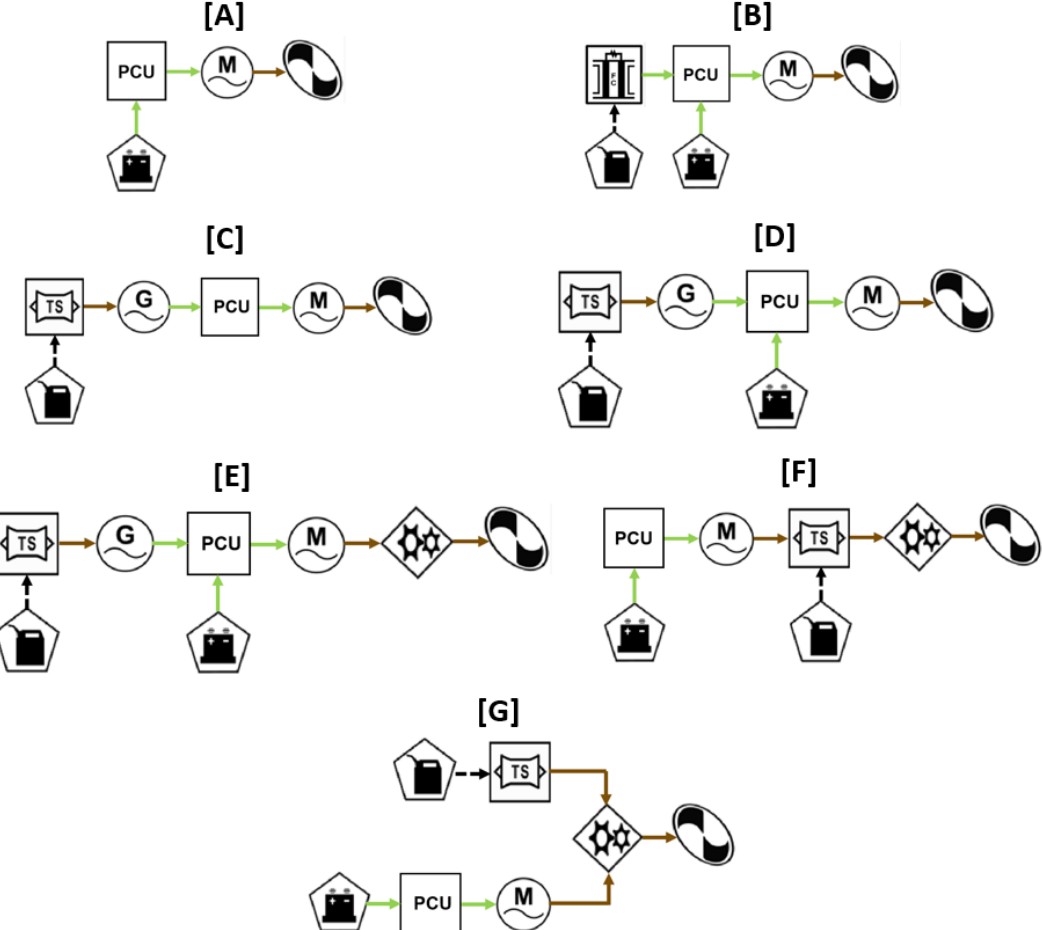

**Figure 6.** Symbolic Representation of Single Channel Propulsion Configurations for Green VTOLs (**A**) All Electric, (**B**) All Electric (Battery and Fuel Cell), (**C**) Turboelectric, (**D**) Series Hybrid Direct Drive, (**E**) Series Hybrid with Gearbox, (**F**) Single Shaft Parallel Hybrid, (**G**) Double Shaft Parallel Hybrid.

At the current state of analysis and using the methodology provided, three configurations are competitive in terms of high reliability and possess moderate to high efficiency potential. All involve quadruplex motors directly driving the propeller and include the improvements projected for motor, generator, and PCU elements of the systems in reliability and efficiency. All electric direct drive configurations, with battery, and with battery and fuel cell, are certainly attractive propositions when used with quadruplex motors. Generally, the literature favors parallel hybrid configurations over series hybrid, e.g., a recent study concluded that parallel hybrid architectures will perform better than series in light, single-engine general-aviation applications with current technologies [75], and series hybrid have also been stated as being less efficient than parallel hybrid configurations [9]. However, multiplex motors were not discussed in any context, and they appear to have been the real improvement factor, along with marginal improvement potential in electrical system efficiencies of course, for series hybrid over parallel hybrid configurations.

## Reliability

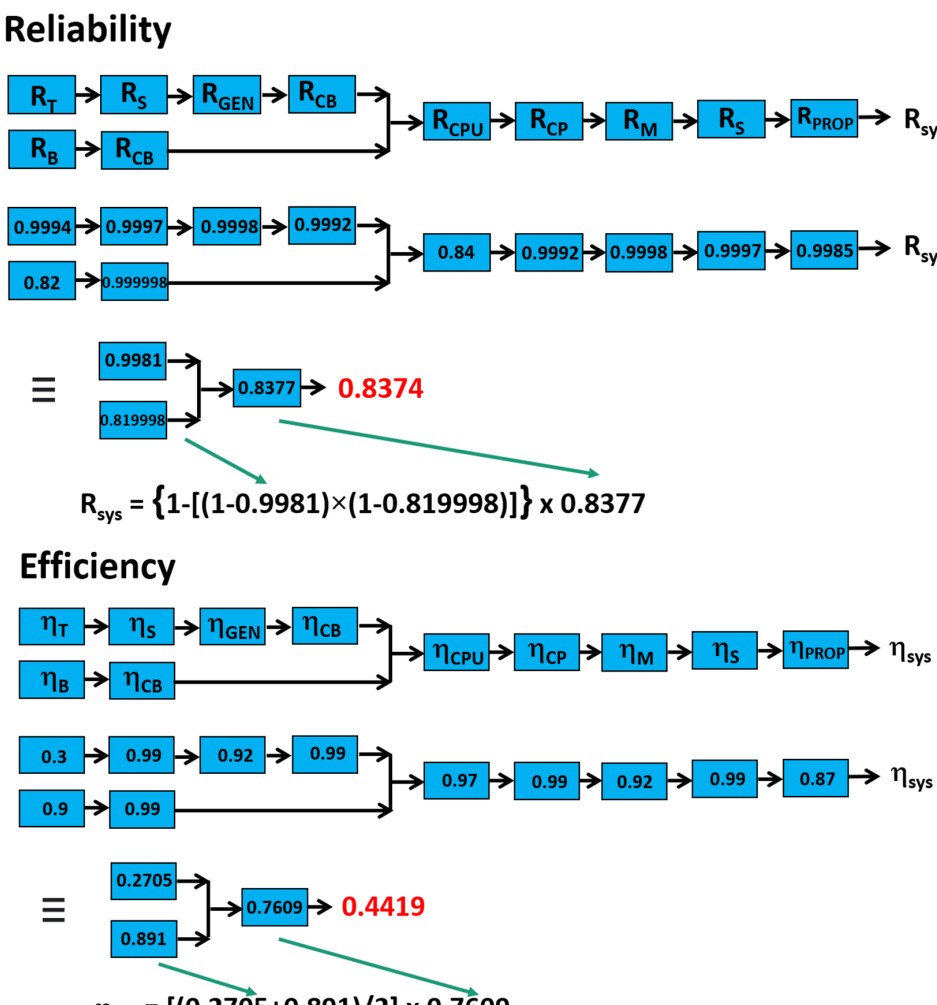

## Efficiency

$$R_{sys} = \{1-[(1-0.9981)\times(1-0.819998)]\} \times 0.8377$$

$$\eta_{sys} = [(0.2705+0.891)/2] \times 0.7609$$

**Figure 7.** Reliability and Efficiency Block Models, Data and Calculations for [D1] Series Hybrid Direct Drive Configuration—Simplex Motor and Current Values.

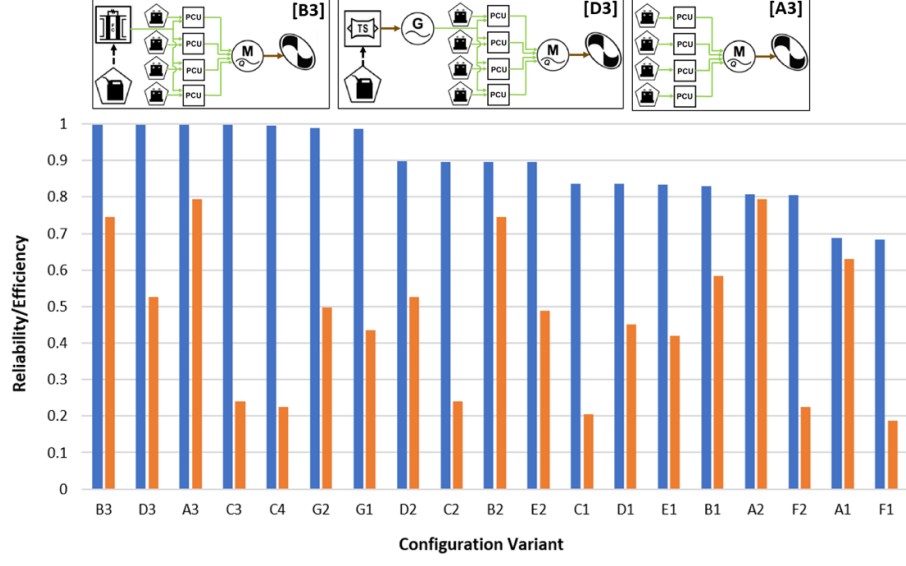

**Figure 8.** Comparison of all Configuration Variants Ranked Left to Right in Terms of Highest Reliability (with symbolic representations for the three leading configurations).

## 4. Case Studies

### 4.1. Electric Tail Rotor (ETR)

Leonardo have already demonstrated the viability of an electric tail rotor [36,79] to replace the traditional complex system of couplings and driveshafts, spanning the distance from the turbine driven main rotor gearbox to the tail rotor gearbox. An electric drive with four independent channels, where each channel is effectively a balanced three-phase motor unit, could reduce often fatal tail rotor gearbox failures [80] and expensive maintenance [81], substantially improving safety and reliability through redundancy. The demonstrator at TRL six proved that a four-channel system works, where the channels intrinsically share the torque demand equally between them. Furthermore, if one channel fails, then the remaining three continue to share evenly the torque demand. The current technology demonstrator, called an 'iron bird rig', is shown in Figure 9.

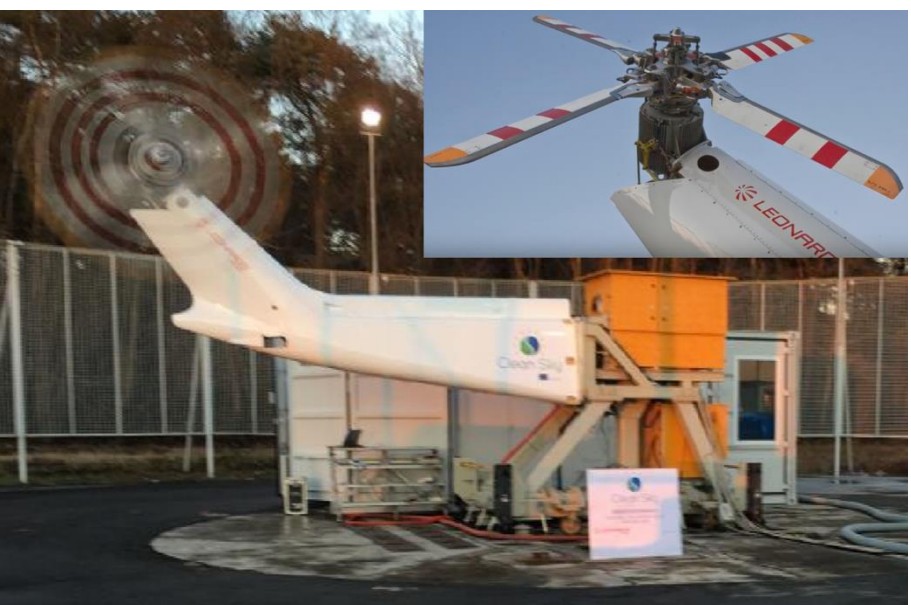

**Figure 9.** Electric Tail Rotor (inset photo showing quadruplex electric drive motor) Undergoing Ground Testing on an Iron Bird Test Rig (Photograph Courtesy of Leonardo).

The study here is concerned with modelling the reliability of the electric tail rotor sub-system when integrated within an existing helicopter platform (what could be called retrofitting) and explore the impact of different multiplex level motors on what could evolve as true hybrid electric propulsion for a helicopter. Although not a new green VTOL concept as such, the case study will also demonstrate further the application of the methodology to a real application. Efficiency is not considered in the study.

The existing helicopter platform has a single output shaft from the main gearbox available to generate power, which provided the mechanical power to the tail rotor originally. Each of the four power channels to the motor requires its own power supply and PCU: each channel providing a quarter of the power for the tail rotor. For the tail rotor thrust to maintain adequate helicopter stability, the loss of just one out of four channels to the quadruplex motor can be tolerated. A parallel arrangement of the generators is desired for maximum redundancy (although series and $2 \times 2$ parallel-series arrangements are also possible). From this information, the symbolic representation of the electric tail rotor can be developed using the adapted AIAA symbols, as shown in Figure 10 (note that the main rotor and shaft from the main gearbox is not shown for clarity, and this model branch is not required to assess the reliability of the electric tail rotor sub-system). The corresponding block models for the overall and simplified equivalent electric tail rotor sub-system is shown in Figure 11.

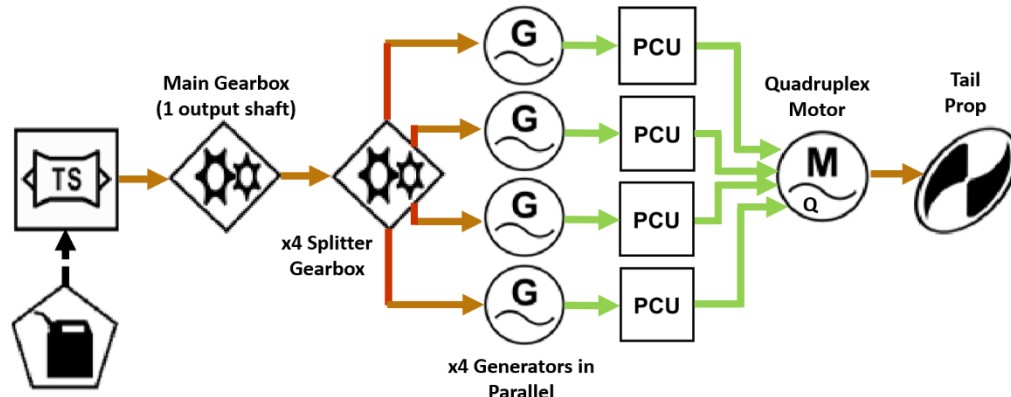

**Figure 10.** Symbolic Representation of the Electric Tail Rotor Retrofitted to an Existing Helicopter.

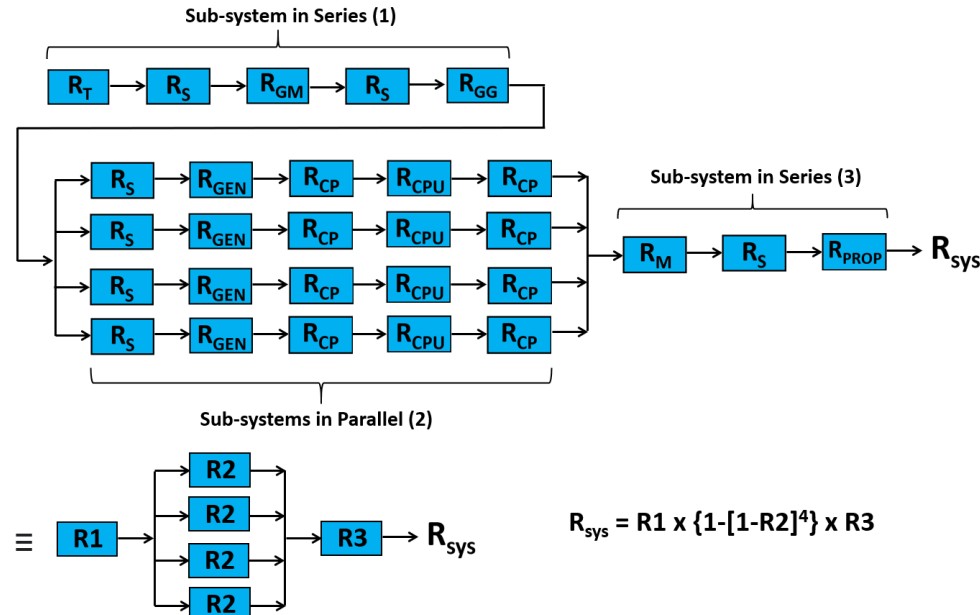

**Figure 11.** Reliability Block Model for the Electric Tail Rotor.

Populating the block diagram with current reliability data for the corresponding components and sub-systems from Table 1, yields a reliability, $R_{sys}$ = 0.991. Even with a quadruplex motor and high redundancy in power channels, the achievement of a target reliability, $R_{sys} \geq 0.9999999$, will not be achievable, mainly due to the series relationship of turboshaft, shafts, and gearboxes, and existing single channel of mechanical power only being available. This demonstrates that the original architecture of the helicopter is not advantageous to an electric tail rotor, unless multiple shafts can be provided from the main gearbox or the four generators are directly driven in some other arrangement from the turboshaft, bypassing both the main and splitter gearboxes.

Further exploration of the multiplex level of the motor can be undertaken, and Figure 12 shows how the reliability falls off with lower levels. The reliability result for the triplex motor is very similar to that for the quadruplex, though having four independent channels is a safety requirement in case one channel fails and reduces the total thrust from the tail rotor, impacting helicopter stability. A supporting study associated with the availability of the electric tail rotor under different arrangements of the four generators also found that the generator and motor reliabilities will need to $R > 0.999999$ to achieve system targets reliabilities [82].

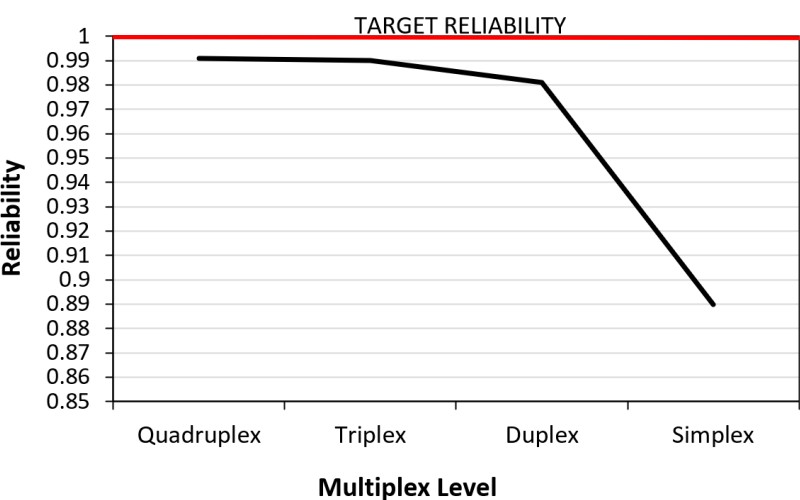

**Figure 12.** Reliability of the Electric Tail Rotor with Different Multiplex Levels.

### 4.2. All Electric VTOL

This case study will investigate the reliability of an all-electric VTOL (eVTOL) with differing number of propulsion channels and multiplex level used for the motors to meet a reliability target. Being all electric, classified [A] from earlier, each motor channel will again be associated with its own PCU and independent channel of power from a battery pack, and the analysis will use the future data set from Table 1 for the reliability of the components and sub-systems specified. Figure 13 shows the symbolic representation of a quad rotor version of the eVTOL using quadruplex motors as an example configuration, though the block model is not shown for conciseness. The study will also include six and eight rotor variants, as well as triplex and duplex motors, both approaches, in combination, providing different levels of redundancy, and therefore reliability of the system.

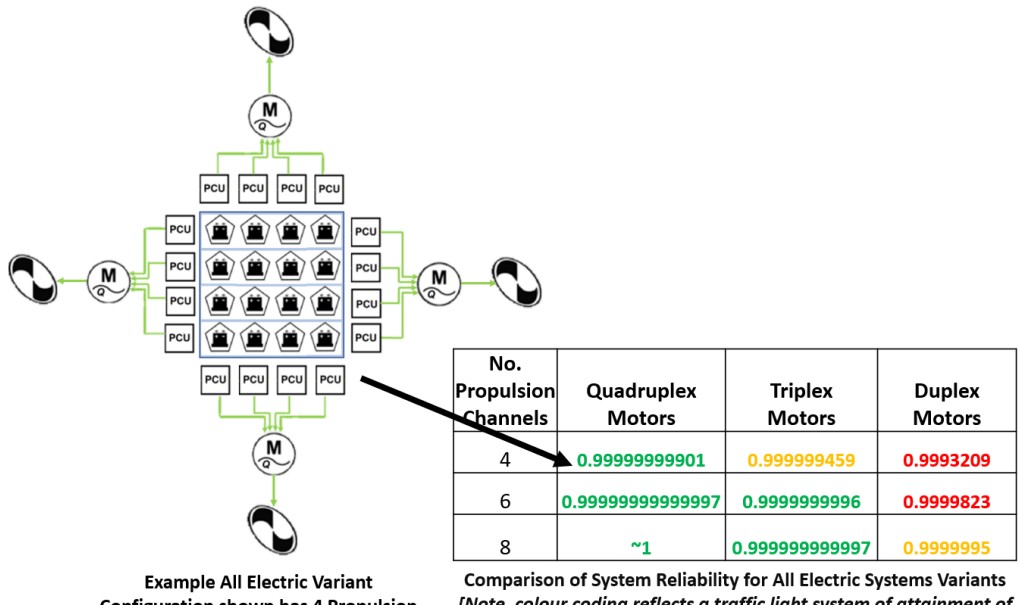

| No. Propulsion Channels | Quadruplex Motors | Triplex Motors | Duplex Motors |
|---|---|---|---|
| 4 | 0.99999999901 | 0.999999459 | 0.9993209 |
| 6 | 0.99999999999997 | 0.9999999996 | 0.9999823 |
| 8 | ~1 | 0.999999999997 | 0.9999995 |

**Example All Electric Variant Configuration shown has 4 Propulsion Channels each with a Quadruplex Motor**

**Comparison of System Reliability for All Electric Systems Variants**
*[Note, colour coding reflects a traffic light system of attainment of the reliability target = 0.9999999]*

**Figure 13.** Reliability for Different Combinations of Propulsion Channel Number and Multiplex Level of Motors to Meet a Target.

The use of multiplex motors for all variants provides the necessary fault tolerance and a degradation of power will occur, not complete loss, if an individual motor stator

winding failure occurs, being a common failure mode in motors [83]. It is assumed that aircraft stability is maintained under this faulted condition, and therefore from Table 3, and FMECA Severity (S) = 4 applies. From the Reliability Map in Figure 5, the target reliability is identified as $R_{sys} \geq 0.9999999$ at Severity (S) = 4.

The analysis of all eVTOL variants is shown in the table on Figure 13. Highlighting the success, or not, of each variant meeting the reliability target is achieved through a traffic light system, where it is evident that the minimum level of multiplex motor for a quad rotor eVTOL is quadruplex, and for six and eight rotor variants, it is triplex, though some margin in confidence in this proposal is being accommodated by using a range about the target reliability, as suggested earlier.

## 5. Conclusions

If green VTOL aircraft are soon to become a reality, then flight safety certification of these new aircraft will be a real challenge for aircraft manufacturers to demonstrate given the variety of technologies, level of maturity, and configurations possible. Several guiding principles can be suggested from the research reported here. Electric propulsion and lift systems can be designed to be fault tolerant and have degraded performance rather than complete loss of performance using multiplex motors. This form of redundancy can be enhanced with redundancy using an appropriate number of propulsion units, though the penalty on mass, cost, and further complexity also needs to be considered. Fault tolerance in electrical motors (and even generators) can be achieved through modularity, multi-channel/phase features to create redundancy, as well as provide tolerance to short and open circuit faults, without a significant increase in machine size, complexity, and mass and should be considered as a primary way of increasing redundancy in aircraft systems to meet reliability targets.

The simplified methodology presented in this paper is a possible starting point for certification purposes, integrating the key requirements of mapping failure modes, failure consequences, and setting appropriate reliability targets. It is intended to be applied to early concepts of green VTOL aircraft, and help visualize these concepts on a common basis, not generate them. There are no limits to the innovation combined with experience that could help generate viable concepts, but this will also be driven by the specification, i.e., mission length, mission type, payload weight, number of people to be carried, level of autonomy, operating environment, speed, etc. To develop the methodology further, the overall power requirements and power distribution though sub-elements is required, derived from mission profiles, to support more detailed trade-off studies, including projections for take-off mass, losses, cost, as well as more detailed reliability and efficiency predictions all becoming possible. Concept designs are usually evaluated relatively against other concept designs, and a successful candidate selected based on many factors in this way. In the approach taken here, reliability and efficiency are the only figures of merit proposed, being the only quantitative measures to help provide decision making at the early stages of design conception that are not related to scale or power distribution factors.

Other related future work is needed to support the concept and more detailed trade-off studies to assess the viability of green VTOLs. The confidence in the reliability and efficiency data used in the studies here is low, as it relies on information supplied to the public domain by operators and suppliers of key components and sub-systems. Up-to-date and representative data (considering environment, loading, and application domain) for the reliability data would be welcomed, though it is acknowledged that this information is commercially sensitive. There is also a great need to understand how aircraft stability for multi-rotor aircraft concepts suffers with complete and partial loss of propulsion and lift capability, as this dramatically changes the consequences of failure and allocation of target reliabilities to minimize risk. Finally, advances in the performance of electrical components and sub-systems such as motors, generators, PCUs, and batteries needs to gather pace, in terms of reliability and efficiency improvements, in order that viable green VTOL architectures can be safely operated.

**Author Contributions:** All authors developed the methodology, J.D.B. and C.P. analysed the case studies and all authors contributed to the writing of the paper. All authors have read and agreed to the published version of the manuscript.

**Funding:** This research was funded by Leonardo: purchase order 4802155000, project title, "Rotorcraft Electric Propulsion Feasibility Study", and ran from October 2019 to September 2020.

**Acknowledgments:** The authors would like to thank engineering staff at Leonardo, Yeovil, including Simon Stacey, Keith Stickels, Chris Charles and Jack Allen, for providing technical input and context for the research conducted in the paper.

**Conflicts of Interest:** The authors declare no conflict of interest.

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
