# Peer review of "Modelling Green VTOL Concept Designs for Reliability and Efficiency"

_designs, 2021_

Round 1

Reviewer 1 Report

Hybrid/electric platforms represent a future goal to reduce the climate footprint of aviation.
The scientific literature seems to be mainly focused on proposing design approaches and workflows, trend studies and optimisations to identify the best solution for a hybrid/electric or full electric aircraft in terms of aerodynamic benefits, weights, direct operating costs, emissions, etc.
If hybrid or full electric aircraft are soon to became a reality, then there is the need to demonstrate the level of maturity, the reliability and the seafety of such platoforms.
This paper fits exaclty this need, focusing on an aspect that must be a key driver in designing a hybrid/electric (or full electric) aicraft. The reliability and the safaty of the conceived system should be taken into account since the eraly stages of aircraft design to produce more detailed trade-off studies and more reliable solutions.
Authors propose an interesting approach to evaluate the reliability and the efficiency of different powerplant arrangements. The methodology presented in this paper represents a good starting point for certification purposes to be applied at early stages of aircraft design. Although, the proposed approach is developed in the context of VTOLS aircraft, its principles are still valid in general for any hybrid/electric (or full electric) flying platforms.
I completely agree with authors when they state that there is the need of more representative data from suppliers and operators concerning key components and sub-systems to produce more and more reliable results.

I suggest the following revisions:

1- You must improve the quality of figures and tables, in some cases they are very hard to read;
2- I suggest to change the color of the arrows representig "Mechanical" or "Electrical" link. You have two different shades of green, it would be hard to appreciate the difference (mainly due to the poor quality of the figures)
3- Check for some typos erros (lines 208-209; 236, 265, 306)
4-Provide a DOI or URL to every entry in the LoR.

Author Response

Please 

Reviewer 2 Report

The paper deals with a reliability study of a VTOL concept design.

The topic is interesting and worthy of pucliation. 

1) The use of a multiplex level propulsion system is obviously more reliable than the simplex one. But for any consideration, it is necessary also to take into account the total weight and the considered power of the motor.

2) In multiplex propulsion, how many "healthy" motors are required for the correct performance of VTOL ? Is the power of each motor equal to the power required by the VTOL?

3) Why the use of a multiphase (>3 phase) electric motor and inverter is not considered?

Author Response

Please find file attached

Round 2

Reviewer 2 Report

No other comments